# Bridging User Perception and Stickiness in Business Microblog Contexts: A Moderated Mediation Model

**Chien-Lung Hsu [1] and Yi-Chuan Liao [2],\*** 

[1]  Department of Business Administration, University of Kang Ning, Tainan 70970, Taiwan; alanhsu8399@gmail.com
[2]  School of Management, Shandong University, Jinan 250100, China
\*  Correspondence: obz703@gmail.com; Tel.: +86-0531-88364664

**Abstract:** This study develops a contingent mediation model to investigate whether user perception enhances customer stickiness through emotional connection and further assess such mediating effect varies with different adaptivity. A moderated mediation approach is adopted to test the hypotheses. Findings reveal the mediating role of emotional connection on the link between perceived usefulness and customer stickiness, but not moderated by adaptivity. On the other hand, the results showed that the relationship between perceived ease of use and customer stickiness is not mediated by emotional connection; however, after considering the moderating effect, our results show that moderated mediation exists.

**Keywords:** perceived usefulness; perceived ease of use; emotional connection; adaptivity; customer stickiness; moderated mediation

## 1. Introduction

Microblog is becoming a popular trend and important social media tool on the Internet [1,2]. For example, Twitter is one of the most well-known microblogs. According to the 2017 annual report of Twitter, there were 330 million active users on Twitter [3]. Another popular microblog is Sina Weibo. It is one of the most popular social media sites in China. In accordance with their financial report of 2017, there were 361 million active users on Weibo. A microblog refers to a set of activities on which persons broadcast brief text updates regarding their daily lives and work, including what they are thinking, reading, and experiencing [4]. Therefore, contemporary firms should consider these new platforms to access their customers. By establishing their own corporate accounts within the context of a microblog, firms may create an effective channel for broadcasting information and connecting with customers. Namely, corporations can post brief text updates about their relevant messages and provide a platform that allows users to discuss and share their experience, thereby helping firms to obtain a better understanding of customer needs and satisfying them. In this respect, corporations utilize that platform to link their customers; and further link their customers and their followers [5]. Consequently, once firms set up their microblog and take advantage of the opportunity of this tool, they may successfully acquire a large number of followers, or even achieve higher rates of customer stickiness [6]. Dell is a well-known example of this phenomenon. Dell publishes a variety of microblog information aimed at different groups, and the different varieties of content are published by different accounts. By positioning their microblogs to address different aspects of their market and brand, Dell can communicate with users worldwide and meet their needs. Users looking for discount information can turn to @DellOut-let. Likewise, when customers are interested in breaking news at Dell, they can turn to @Direct2Dell, and so on. Via the microblogs, Dell stays in touch with microblog users in many countries, including the United States, Brazil, Mexico, China, and Japan.

Microblogs have been the subject of many recent academic studies; yet, there are still some unanswered questions regarding the effect of user perception on customer stickiness within the microblog context. Based on the theory of reasoned action (TRA) and technology acceptance model (TAM), perceived usefulness and perceived ease of use are the two important factors that can predict the behavioral intention to use technology [7]. These two factors have been widely applied in the study of information system adoption and used in online marketing management [8,9]. However, perceived usefulness and perceived ease of use may not directly reveal all the motivations of microblog users [10]. In a modern purchasing process, online shopping is evolving as a more relational-based exchange, which emphasizes user's experience and personal interaction in their online shopping communities [11]. This study, therefore, uncovers the intervening factor affecting the relationship between user perception and customer stickiness, and argues that emotional connection plays an intermediate role in this relationship. Emotional connection has been identified as a key driver of the success of a virtual community. The term "emotional connection" refers to the bonds developed over time through positive interaction with other members. As customers interact with other members, they are more likely to build a connection and come to view themselves as members of a certain group, such as brand community and consumption community [12,13]. Study findings indicate a relationship between emotional connection and the organizational performance of a microblog website [14–16]. However, little is understood of the mediating effect between user perception (i.e., perceived usefulness and perceived ease of use) and the customer stickiness of the microblog. Accordingly, an interesting question arises: Does emotional connection mediate the relationship between user perceptions and customer stickiness?

In addition, a firm's superior performance depends upon the appropriate alignment of its strategic actions with different situations [17]. Based on this logic, this study proposes the use of adaptivity (system), which enables corporations to adjust themselves for users through obtaining information and facilitating suitable modifications to the user interface [18]. Adaptivity allows firms with microblogs to effectively and efficiently customize messages that more precisely match products to customer needs, thereby resulting in customer satisfaction and stickiness [19]. Therefore, another interesting question arises: Does adaptivity affect the mediating effect of emotional connection within the microblogs? Put differently, is the relationship between user perceptions and stickiness through emotional connection contingent on adaptivity?

This study views these enterprise microblogs as a type of virtual brand community in which corporations disseminate brief messages and offer platforms for individuals to communicate, share information and experiences, and interact with other members. This study extends the existing literature in two respects. First, whereas previous studies have typically examined either the moderating or the mediating effects in the microblog context [4,6], this study empirically tests the relationships which incorporate both moderation and mediation into a single model. The current study illustrates how emotional connection mediates the relationship between user perception and customer stickiness, and how the mediating process varies as a result of the functionality of adaptivity. Second, this study contributes to the online human behavior theory by showing how emotional connection plays a crucial mediating role by creating a sense of belonging and mutual history, thus creating customer value in the virtual group of the microblog. The paper further discusses the use of adaptivity to generate different mediating processes, which emphasizes the requirement for customized messages to fit customer needs and deliver more valuable information for users. Here are brief highlights of our contribution:

- This study proposes that emotional connection mediates the link between user perception and customer stickiness.
- This study develops the moderated mediation model to argue that the mediating effect of emotional connection relies on the adaptively system.

The remainder of this paper is organized as follows. Section 2 presents the literature review and develops the research hypotheses, as guided by the theoretical model. Section 3 addresses the research methods. Section 4 reports the results. Finally, Section 5 concludes the study.

## 2. Literature Review and Research Hypotheses

Figure 1 presents the theoretical framework investigated in this study. This framework shows that emotional connection mediates the relationship between user perception and customer stickiness. In addition, this study illustrates how adaptivity is a moderating variable and how the mediating process varies as a function of adaptivity. The following sections detail the specific arguments regarding these relationships and present the underlying rationale.

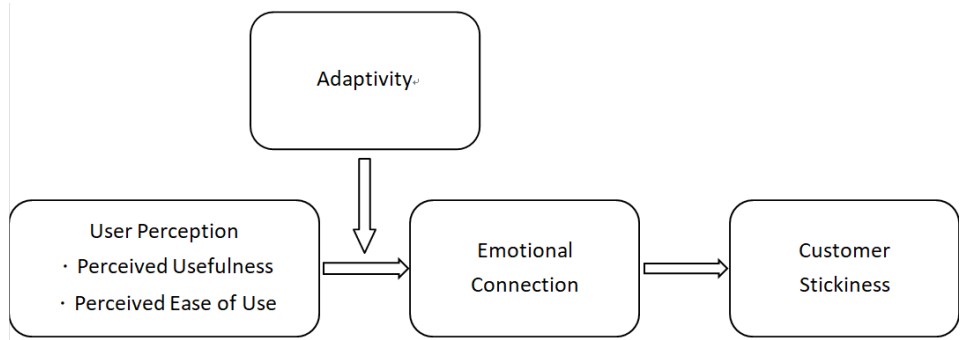

**Figure 1.** Theoretical framework.

### 2.1. Literature Review

#### 2.1.1. Emotional Connection

The microblog is a type of virtual community where people meet to discuss common interests. Users become attached and often return to their communities [20], sometimes becoming so dependent upon the community that such users can be described as being addicted to it [21,22]. Because virtual community has its own history, it can be regarded as a community of memory. Although, a community is characterized mainly by the relational interactions or social ties that draw people together, communities are not automatically formed. Particularly, users in microblogs communicate with each other and exchange information in the form of instant messages. In this way, communities are not merely about aggregating information or resources, but about attracting members and bringing them together to meet some of their social and commercial needs [23]. As such, emotional connection in microblogs might be the crucial factor influencing the virtual community's success [16]. Emotional connection is a feeling that one has invested part of oneself to become a member and interact with other members regarding mutual and specific issues, thereby strengthening the feeling of belonging [24,25]. This feeling of togetherness and belonging is accompanied by the notion of being part of a group, spending time together, companionship, socializing, and networking. According to some social psychology studies, humans need to belong to a group to be affiliated with others [12,26] because groups provide individuals with a source of information and help in achieving goals as well as giving rewards [27].

#### 2.1.2. User Perception

In accordance with the TRA, an individual's perception of society's attitude toward a behavior is the main predictor of behavioral intention, which, in turn, is the primary determinant of actual behavior [28]. The TAM also explains that user acceptance of a technology depends on user perception [7], which helps to predict individuals' acceptance of an innovative technology. Specifically, TAM identifies the causal relationships that exist among individuals' perception of a technology's usefulness and ease of use, and their behavioral intention to use the technology. Perceived usefulness refers to the degree to which a person believes that using a particular system enhances his or her job performance [7]. A useful system encourages more exploration and greater participation in the system [16]. In the context of this study, perceived usefulness refers to the users' belief in their ability to obtain information

and share their experiences with others while using the microblogs. This implies that microblogs with useful information will show an increase in sharing behavior and developing emotional connection. Perceived ease of use refers to the degree to which a person believes that using a particular system would be free of effort [7]. A system with increased perceived ease of use will encourage users to try the system and participate frequently [29]. That is, users expect they will not need to spend significant time and effort to operate a system. From these definitions, it becomes clear that perceived ease of use is important to ensure the system's attractiveness to users, especially since usability has been found to be positively associated with increased system usage [29–31]. Therefore, this study expects that individuals' perceived ease of use of a system enabling an online community positively influences users to participate in and devote to that community.

### 2.1.3. Adaptivity

Adaptivity is the ability of systems to adapt the rules, structure, and information content of a virtual community to meet user needs [16]. Adaptivity includes changing the interface design and features, such as the provision of search facilities, tracking and navigation aids, and the management of favorite topics [32]. An adaptive system also encompasses the creation and customization of a personal website and its content. Prior study [33] stated that for a social system to improve its outcomes, its structures should be adapted continuously but in a stable manner to meet the needs of the tasks and the group. In general, since it is not possible to meet every user's needs within a single virtual community, adaptivity may play an important role in influencing the characteristics of a virtual community, including its success or failure [15,16].

### 2.2. Mediating Effect of Emotional Connection on User Perception and Customer Stickiness

Previous studies that have examined information system usage from a longitudinal perspective have found that system usefulness is a critical perception that drives the intention to continue using the system [27,29,34]. Within the context of the microblog, perceived usefulness representing the utility of an online transaction indicates that users have a certain ability to acquire useful information and services, share their experiences with others, and enhance their performance while using the microblog [35]. In our study, perceived usefulness is based on this perspective and refers to the degree to which individuals perceive the advantages of using the microblog. One of the main reasons for individuals to use the media is to fulfill their cognitive needs such as to acquire information, knowledge, and understanding [36]. A useful platform can cause users to be more willing to participate and search for more information to fulfill their needs, and therefore, become conscious of a community within the enterprise microblogs. This sense of a virtual community is a feeling of belonging and attachment toward a community on the Internet [37,38]. Specifically, users' perception of the benefits of the microblog is likely to lead to a stronger intention to be more deeply involved and interact with others via the microblog, increasing the rate of emotional connection. During these processes, individuals also easily perceive the support of other members, thereby deepening the relationship between users and their community [39]. Furthermore, the more the members interact with these communities, the higher the quality of information they perceive, which are derived from prior positive interactive experience and developed community memory [40]. The superior quality of the information causes the members to consider virtual community as a credible information source, which increases their commitment to spend more time on the microblog.

In sum, increased perceived usefulness induces users to access information via the microblog, which provides a platform to discuss and gain solutions. The interactive processes can fulfill the users' cognitive needs and strengthen their social connections, resulting in an increase in users' bond. Furthermore, increased emotional connection can help to build trust, therefore enhancing the likelihood that the information will be perceived as being of high quality. This, in turn, facilitates user commitment and leads to an increase in users' intention to remain connected with the virtual community. As such, our hypothesis is as follows:

**Hypothesis 1 (H1).** *Perceived usefulness has an effect on customer stickiness through emotional connection.*

Perceived ease of use refers to the users' belief that the microblog is easy to operate. Since effort is a finite resource, applications that are perceived to be easier to use than their competitors are more likely to be accepted by users [7,8]. The microblog is a place where users can share interests and resources, engage in joint activities, and work toward the same goals. As microblogs are easier to use, such platforms encourage individuals to participate and share important information [41]. During these processes, users increase their interactions with other members, which results in a higher sense of belonging [16]. Specifically, as ease of use increases, users can more easily chat together, exchange information, and deepen their relationships and connections. In fact, user behavior in the microblog is intertwined, in which members' roles and identities change gradually from participator to core contributor, encouraging them to share more information [42]. The changes in members' identities, indicating how they perceive themselves and how other members think about them, is a result of their participation and engagement in online interactions. Once members develop a sense of belonging to the group, they share their faith with other members in the group, and consequently, their needs will be met through their commitment to each other over the long term. Prior study [43] pointed out that intensive member interactions will strengthen the bonds among members and result in a stronger social network [44], which can enhance user lock-in [45]. Therefore, as the perception of ease of use increases, users are induced to exchange information via the microblog, and help other users to develop their connection through the interaction. Furthermore, the increase in connection can facilitate an increase in members' commitment to each other, leading them to stay longer within the microblog. As such, perceived ease of use can prompt customer stickiness by increasing the impact of emotional connection.

**Hypothesis 2 (H2).** *Perceived ease of use has an effect on customer stickiness through emotional connection.*

*2.3. Moderating Role of Adaptivity on the Mediating Effect of Emotional Connection*

From the market orientation theory, it is clear that analyzing the market situation creates opportunities for firms to invest in products or services that satisfy unmet needs [46]. Market orientation signals firms to work on certain problems. Therefore, firms with higher market-oriented propensity are likely to access more information and knowledge about customers' preference. In microblogs context, corporations could use an adaptive to satisfy online customers' needs. Adaptivity refers to the ability of systems to adapt themselves to the user by acquiring information and then triggering suitable modifications to the user interface. Corporations operating adaptivity could offer personalized content or advertisements based on the user's profile and interests [47,48]. By customizing some features of a product or service, firms can allow a user to enjoy more convenience, lower cost, or some other benefits [10]. With such a wide diversity of available information, and a wide variety of interests among users of the microblog, it is important that adaptive features are incorporated by firms to change system parameters to suit users' preferences. An intelligent system with an adaptive interface can keep track of user activities and profiles so as to dynamically generate unique services and information [49]. As such, adaptivity has an important impact on the microblog, thus, enhancing the success of firms.

As users perceive the usefulness of a microblog and increasingly interact with others to gain better information and solutions, intensive connections arise and users are further willing to be involved with the website, resulting in staying longer on the website. Thus, this study argues that members' perceived usefulness of the microblog has a positive impact on customer stickiness through the emotional connection construct. This study further argues that the mediating process varies as a direct result of the system's adaptive functions. Specifically, mediating processes enable users to access diverse information and messages but may bring an excessive amount of information that receivers cannot process efficiently [4]. As individuals have bounded rationality, excessive information and messages can distract them and cause stress, thus, decreasing individuals' motivation to interact with

others and reducing their time on the website. To deal with these problems, firms can use adaptive to provide personalized information, which can reduce the perceived complexity that was caused by excessive information, thereby reducing customers' search efforts and increasing user satisfaction [50]. Moreover, adaptivity is inherently preferred, since it takes individual needs into account and creates a relationship of trust between the website and the user [15]. Therefore, when corporations use an adaptive to satisfy users' needs and provide precisely the information and solutions that suit those users, the users develop more connection with their members, which, in turn, becomes more willing to remain on the website. This study expects that the higher the level of adaptivity, the more positive the mediating effect of emotional connection on the perceived usefulness–customer stickiness relationship.

**Hypothesis 3 (H3).** *The mediating effect of emotional connection on the link between perceived usefulness and customer stickiness depends on level of adaptivity.*

This study argues that increasing the level of perceived ease of use would positively impact customer stickiness via the emotional connection. This study further proposes that the mediating process varies as a result of the adaptivity functionality. Specifically, a high level of adaptivity by corporations enables users to access more precise and concise information based on the member identity process; therefore, members sense more benefits from perception of ease of use and generate more positive feelings [32], which, in turn, produce a higher level of customer stickiness. In other words, when the microblog system adapts to the needs of the individual, access to system services and information are made more efficient and effective for that individual, freeing him or her of the effort to perform repetitive or otherwise time-consuming tasks when using the system [16]. Therefore, when corporations use an adaptive system to accelerate the process of information exchange, users are more likely to share information with the community and increase their bonding with others, therefore, enhancing users' willingness to remain on the website. On the contrary, a low level of adaptivity impedes firms to align messages or information with the needs of users, and may, therefore, consume more time of users on irrelevant issues or topics, even if they perceive the system to be easy to use. As such, the mediating process will have a less positive impact when the level of adaptivity is low than when the adaptivity is high.

**Hypothesis 4 (H4).** *The mediating effect of emotional connection on the link between perceived ease of use and customer stickiness depends on level of adaptivity.*

## 3. Research Method

### 3.1. The Moderated Mediation Model

In accordance with previous study [51], the moderated mediation model was used to examine the above hypotheses. The model combines moderation and mediation simultaneously, which is termed as conditional indirect effect or conditional process modeling. Such a mechanism linking the independent variable to the dependent variable can be conditional if the indirect effect of the former on the latter through a mediator is contingent on a moderator. There are many different ways in which this could happen [52]. As such, in line with our hypotheses, the equations are as follows:

$$EC = a_0 + a_1PU + a_2PEU + a_3Apt + a_4PU * Apt + a_5PEU * Apt + X'B$$
$$St = b_0 + b_1PU + b_2PEU + b_3Apt + b_4EC + b_5PU * Apt + b_6PEU * Apt + X'B$$
$$\text{Conditional indirect effect on PU equation}: (a_1 + a_4CA) * b_4$$
$$\text{Conditional indirect effect on PEU equation}: (a_2 + a_5CA) * b_4.$$

Note:

EC: Emotional Connection. PU: Perceived Usefulness. PEU: Perceived Ease of Use.
St: Stickiness. Apt: Adaptivity. X'B: A linear combination of the other variables.

### 3.2. Data Collection and Measures

The data used in this research were collected through an online sampling survey. The respondents who had experience using a microblog webpage of a corporation were selected. On the cover page of the questionnaires, this study ensured respondents that the collected information is only for academic use, and declared the voluntary nature of this survey and the confidentiality to every participant. To encourage participation, the respondents were offered the opportunity to win a lottery-based prize (e.g., 7-11 coupons, familymart coupons). The respondents were asked to evaluate the items in a questionnaire based on their usage behavior on the microblog. All items were measured on a seven-point Likert scale ranging from 1 ("strongly disagree") to 7 ("strongly agree"). After discarding the invalid responses, this study conducted the statistical analysis.

To examine the hypothesis, the paper specifies the definition and measurement of each variable. This study revised the popular work [10], which defined perceived usefulness as the extent to which a person believes that using a particular system will enhance performance toward the goals, and perceived ease of use as the degree to which a person believes that using a particular system would be free of effort. This study measured perceived usefulness with five items and perceived ease of use with three items. Based on the work of prior study [4], this study defined emotional connection as bonds developed over time through positive interactions with other members, and used five items to measure the construct. Furthermore, based on the study [16], adaptivity was defined as the ability of systems to adapt rules, structures, and information contents of a virtual community to meet user needs, and was measured using three items. In line with the study [53], customer stickiness was defined as the ability of a website to attract and retain visitors, and was measured using four items.

In addition, this study controlled for sex, age, and spending time on the internet of respondents in order to reduce their potentially confounding effects on emotional connection and stickiness. Research indicates that younger people tend to adopt the online community faster (e.g., [42]). Females may be more likely to be involved in a virtual community and stick to a website [27]. Spending time referred to the amount of time users spend on the Internet every day. Scholars suggest that as individuals stay longer on the Internet, they are more likely to be familiar with the firms' websites and easily stay [54].

### 3.3. Assessing the Reliability and Validity of Measures

To test the construct validity, this study employed the confirmatory factor analysis (see Appendix A). The results showed that the overall model provides a satisfactory fit to the data ($\chi^2$/d.f = 506.42/109 = 4.48, CFI = 0.97, IFI = 0.97, RFI = 0.96). All item loadings were significantly on their expected constructs ($p < 0.01$). Composite reliability (CR) for the five constructs exceeded 0.70. This means appropriate convergent validity and reliability [55,56]. With regard to discriminant validity, this study deployed several techniques. First, this study examined whether the square root of the AVE of each variable was greater than the correlation between construct variables (see Table 1). The results show emotional connection had slightly weak discriminant validity. Further, this study ran a series of chi-square tests for all constructs in pairs to determine whether the unconstrained model is significantly better than the constrained model [57]. All combinations resulted in a higher critical value ($\Delta\chi^2(1)$ = 3.84 at the 5% significance level), indicating acceptable discriminant validity for each scale. Moreover, this study, in accordance with the work [57], determines the confidence internal (±two standard errors) around the correlation estimate between any two constructs included a value of one. Our results indicated that none of the confidence intervals of all pairs of constructs included a value of one. Although the first technique showed that emotional connection had marginally weak discriminant validity, this study used the series statistics method to check this, and the results show that discriminant validity is not a problem in our study. Thus, this study concluded that our measures were valid. Table 1 presents the standard deviations of means and the correlations of the constructs used in the following analyses.

**Table 1.** Correlation Matrix and Descriptive Statistic Measures.

|      | PEU   | PU    | Apt   | EC    | St    | Sex  | Age  | Edu  | Time |
|------|-------|-------|-------|-------|-------|------|------|------|------|
| 1    | 0.70  |       |       |       |       |      |      |      |      |
| 2    | 0.61  | 0.67  |       |       |       |      |      |      |      |
| 3    | 0.45  | 0.46  | 0.54  |       |       |      |      |      |      |
| 4    | 0.44  | 0.66  | 0.50  | 0.64  |       |      |      |      |      |
| 5    | 0.38  | 0.57  | 0.38  | 0.66  | 0.66  |      |      |      |      |
| Mean | 16.16 | 26.46 | 14.04 | 24.05 | 25.27 | 0.39 | 2.48 | 5.21 | 1.70 |
| SD   | 2.86  | 4.69  | 2.97  | 4.37  | 4.89  | 0.49 | 0.90 | 0.66 | 0.85 |

NOTE: The diagonal elements are square roots of the AVE. The lower-left triangle elements are correlations among the composite measures; [2] PEU: Perceived ease of use. PU: Perceived usefulness. Apt: Adaptivity. EC: Emotional connection. St: Stickiness. Edu: Education level; time: Spend-Time on Internet.

As our data were collected from a single source, the common method variance should be considered. Therefore, this study conducted a series procedure to minimize the common method bias. The study assured each participant that their responses would be anonymous and there were no right or wrong answers to the questions. Moreover, the study also employed statistical methods to examine the common method bias that would not pose any problem in our analysis [58]. First, this study conducted the well-known Harman's one-factor test. Specifically, this study employed the exploratory factor analysis for all our constructs. The results indicated that no substantial common method bias existed in the data [59]. Second, based on the research (e.g., [60,61]), this study tested for common method bias by applying a confirmatory factor approach. The results showed that no single latent factor accounted for all manifest variables. Overall, these statistical evaluations suggested that the common method bias did not affect this study.

## 4. Results

This study verified the mediation models mentioned earlier using the indirect effects approach. The indirect effect of independent variables on criterion variables through the mediator can be quantified as the product of paths [52,62]. The results from Table 2 denote our mediated effects; Model 1 represents the effects of perceived usefulness and perceived ease of use on emotional connection ($\beta$ = 0.59 and $\beta$ = 0.09) and Model 2 indicates the effects of emotional connection on stickiness ($\beta$ = 0.59). According to the product-of-coefficients approach, this study computes the ratios of product term to estimate standard error. The results suggest that the mediating effect of emotional connection from perceived usefulness to stickiness is significant ($\beta$ = 0.35, $p$ < 0.01), and that the mediating effect from perceived ease of use to stickiness is insignificant ($\beta$ = 0.06, $p$ > 0.05); therefore, H1 is supported, but H2 is not supported.

**Table 2.** Mediation analysis (log likelihood= -7971.85, estimation method = ML).

| Independent Variable | Criterion: Emotion Connection | | | Criterion: Stickiness | | |
|---|---|---|---|---|---|---|
| | Model 1 | | | Model 2 | | |
| | $\beta$ | Z Value | *p*-Value | $\beta$ | Z Value | *p*-Value |
| Sex  | 0.99  | 3.37 ** | 0.001   | −0.37  | −1.15    | 0.25   |
| Age  | 0.22  | 1.38    | 0.17    | 0.07   | 0.41     | 0.68   |
| Edu  | 0.001 | 0.01    | 0.99    | −0.002 | −0.01    | 0.99   |
| time | 0.20  | 1.21    | 0.26    | −0.04  | −0.02    | 0.84   |
| PU   | 0.59  | 15.70 ** | <0.001 | 0.21   | 4.22 **  | <0.001 |
| PEU  | 0.09  | 1.51    | 0.067   | 0.05   | 0.07     | 0.24   |
| EC   |       |         |         | 0.59   | 12.62 ** | <0.001 |
| Indirect Effects on stickiness through emotion connection | | | | | | |
| perceived usefulness ->stickiness | | | 0.35 | 9.84 ** (*p* < 0.01) | | |
| perceived ease of use ->stickiness | | | 0.06 | 1.50 (*p* = 0.068) | | |

NOTE1: N = 543. * *p* < 0.05, ** *p* < 0.01; NOTE2: This study used a two-tailed test for control variables and a one-tailed test for all hypotheses. Edu = education level; time: Spend-Time on Internet; PU; Perceived Usefulness; PEU: Perceived Ease of Use; EC = emotional connection.

This study further examines how adaptivity moderates the mediating effects of emotional connection on the user perception–stickiness relationship. The statistical results are reported in Tables 3–5. In Table 3, Model 3 shows the coefficient estimate of perceived usefulness and perceived ease of use on emotional connection ($\beta = 0.63$, $p < 0.01$ and $\beta = -0.38$, $p < 0.05$), and also that of the product term of independent variables and moderator ($\beta = -0.01$, $p > 0.05$ and $\beta = 0.03$, $p < 0.05$). Model 4 presents the coefficient estimate of emotional connection on stickiness ($\beta = 0.58$, $p < 0.01$). In the moderated mediation model, the interaction coefficient should depart from zero [63]. Based on this perspective, adaptivity may not moderate the mediating effect of emotional connection from perceived usefulness to stickiness, but adaptivity has a significant indirect effect of perceived ease of use on stickiness through emotional connection. However, merely inspecting the signs and magnitudes of regression coefficients is generally insufficient analysis for moderated mediation hypotheses [52]. Thus, to gain insight into the moderated mediation process, this study examined the nature of the conditional indirect effects using the procedure suggested by the study [51]. In Table 4, the procedure tests the significance of regression-coefficient estimates for the relation of perceived usefulness to stickiness via emotional connection at the standard deviation above and below the mean of adaptivity. A significantly positive relationship was found ($\beta = 0.30$, $p < 0.01$) at a mean level of adaptivity. At high levels of adaptivity, the indirect effects of perceived usefulness on stickiness are significantly positive ($\beta = 0.28$, $p < 0.01$ and $\beta = 0.27$, $p < 0.01$ for high and very high levels). At low levels of adaptivity, the indirect effects of perceived usefulness on stickiness are positively significant ($\beta = 0.31$, $p < 0.01$ and $\beta = 0.33$, $p < 0.01$ for low and very low levels). This implies that as adaptivity increases, the indirect effect decreases, but the difference (i.e., $\beta = 0.33$ vs. 0.27 for very low vs. very high levels, respectively) in the effect of moderated mediation was not significant ($Z = 0.88$, $p > 0.05$). In sum, adaptivity slightly mitigates the mediating effect of emotional connection, and thus, H3 is not supported. Figure 2 shows the effect of moderated mediation (MOME) and illustrates that the slope was not significantly different.

Once again, for Hypothesis 4, this study verified the indirect effect of perceived ease of use on stickiness at different values of adaptivity. Table 5 denoted the conditional effect of adaptivity on the mediating effects of emotional connection on the relationship from perceived ease of use to stickiness. This study found that as stickiness increases, the indirect effects increase and are significantly positive at a very high level of adaptivity ($\beta = 0.13$, $p < 0.05$). The results indicated that a high level of adaptivity strengthens the mediating effects of emotional connection. Therefore, adaptivity has positive impacts on the mediating effects of emotional connection, thus supporting H4. Figure 3 presents the effect of this moderated mediation and finds that the slope was significantly different.

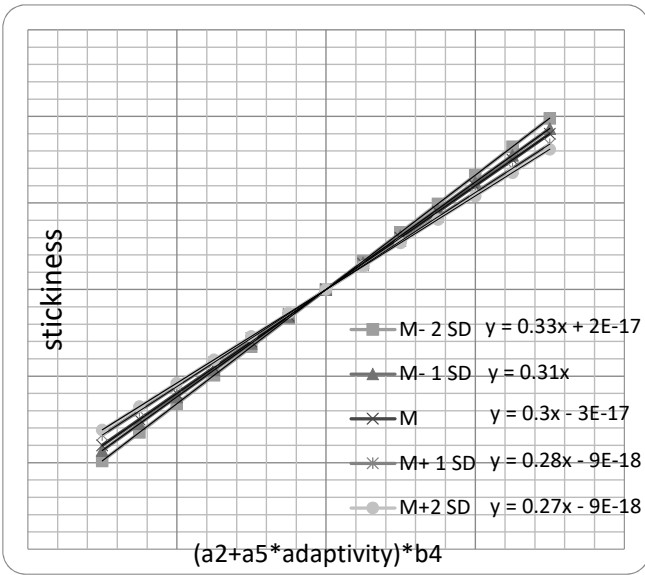

**Figure 2.** The effect of moderated mediation (perceived usefulness -> emotional connection -> stickiness).

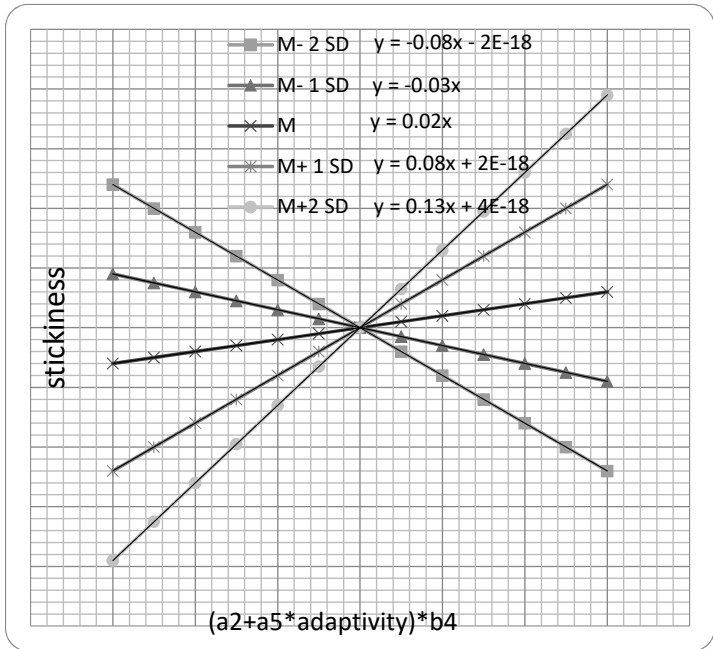

**Figure 3.** The effect of moderated mediation (perceived ease of use -> emotional connection -> stickiness).

**Table 3.** Moderated mediation analysis (log likelihood=$-13{,}544.484$, estimation method = ML).

| Independent Variable | Criterion: Community Connection | | | Criterion: Stickiness | | |
|---|---|---|---|---|---|---|
| | **Model 3** | | | **Model 4** | | |
| | β | Z value | *p*-value | β | Z value | *p*-value |
| Sex | 0.82 | 2.89 ** | 0.004 | −0.36 | −1.13 | 0.26 |
| Age | 0.11 | 0.75 | 0.45 | 0.05 | 0.30 | 0.76 |
| Edu | 0.03 | 0.15 | 0.88 | 0.002 | 0.01 | 0.99 |
| time | 0.22 | 1.36 | 0.174 | −0.03 | −0.15 | 0.88 |
| PU | 0.63 | 4.56 ** | $p < 0.001$ | 0.47 | 2.93 ** | 0.002 |
| PEU | −0.38 | −1.91 * | 0.29 | −0.16 | −0.71 | 0.24 |
| Apt | 0.10 | 0.50 | 0.62 | 0.32 | 1.41 | 0.16 |
| PU * Apt | −0.01 | −0.88 | | −0.02 | −1.73 * | 0.04 |
| PEU * Apt | 0.03 | 2.00 * | | 0.01 | 0.86 | 0.19 |
| EC | | | | 0.58 | 11.92 ** | <0.001 |

NOTE1: N = 543. * $p < 0.05$, ** $p < 0.01$; NOTE2: Edu = education level; time: Spend-Time on Internet; PU: Perceived usefulness; PEU: Perceived ease of use; Apt: Adaptivity; EC = emotional connection; NOTE3: This study used a two-tailed test for control variables and a one-tailed test for all hypotheses.

**Table 4.** Conditional effects at adaptivity (mean and ±2 SD) on PU->EC-> Stickiness.

| Adaptivity Values | β | SE | Z | *p*-Value |
|---|---|---|---|---|
| M- 2 SD | 0.33 | 0.046 | 6.97 ** | $p < 0.001$ |
| M- 1 SD | 0.31 | 0.036 | 8.46 ** | $p < 0.001$ |
| M | 0.30 | 0.033 | 8.93 ** | $p < 0.001$ |
| M+ 1 SD | 0.28 | 0.037 | 7.43 ** | $p < 0.001$ |
| M+2 SD | 0.27 | 0.05 | 5.49 ** | $p < 0.001$ |

NOTE1: * $p < 0.05$, ** $p < 0.01$, β = (a1 + a4*Adaptivity)*b4; NOTE2: This study used a two-tailed test for control variables and a one-tailed test for all hypotheses.

**Table 5.** Conditional effects at adaptivity (mean and ±2 SD) on PEU->CC-> Stickiness.

| Adaptivity Values | β | SE | Z | *p*-Value |
|---|---|---|---|---|
| M- 2 SD | −0.08 | 0.053 | −1.47 | 0.071 |
| M- 1 SD | −0.03 | 0.038 | −0.71 | 0.024 |
| M | 0.02 | 0.038 | 0.65 | 0.026 |
| M+ 1 SD | 0.08 | 0.053 | 1.44 | 0.075 |
| M+2 SD | 0.13 | 0.07 | 1.86 * | 0.042 |

NOTE1: * $p < 0.05$, ** $p < 0.01$, β= (a2 + a5*adaptivity)*b4; NOTE2: This study used a two-tailed test for control variables and a one-tailed test for all hypotheses.

## 5. Conclusions

### 5.1. Discussion

This study presented a moderated mediation model to discuss the relationship between user perception and customer stickiness. First, this paper argues that emotional connection plays an important role in mediating the link between them. Second, this study proposes that adaptivity can moderate the mediating effect of emotional connection. The results showed that the relationship between perceived usefulness and customer stickiness can be mediated by emotional connection, but that mediating effect cannot be moderated by adaptivity. That is, when customers perceived more usefulness of the microblog, it can increase the feeling of emotional connection with other members, thereby resulting in increasing the customer stickiness of the microblog. However, adaptivity does not impact the mediating process. One possible reason for this result is that community members sense the usefulness of microblog for access of information that allows members to satisfy their needs, and that microblog offers a platform to develop their social connection, therefore strengthening their commitment to stay in the virtual community. Thus, the sense of community effect is aroused without adaptive systems to fit their needs [4]. Specifically, during this psychological mechanism, they could only devote little cognitive effort in absorbing what members receive from the microblog, and then they fall into the state of flow [29,64]. As such, adaptivity is not necessarily applicable for those members who build the sense of community.

Furthermore, the results showed that the relationship between perceived ease of use and customer stickiness is not mediated by emotional connection; however, after considering the moderating effect, the findings indicated that moderated mediation exists. That is, the mediating effect of emotional connection was contingent on adaptivity. Specifically, corporations determined that the ease of use of the system would not develop an emotional connection with online users, and consequently, would not attract users to stay longer. Due to time and capability limitations, large amounts of messages may cause information overload, make users confused, and lead to poor decisions. Adaptivity provides the solution to resolve these problems. Under high levels of adaptivity, users can access more precise and reliable information. In such a situation, when users sense an easy platform to operate, they may increase their involvement and interactions with other people, thereby developing an emotional connection that, in turn, strengthens their willingness to stay. Therefore, firms should notice that the ease of use of a microblog is a necessary, but not sufficient, condition. After incorporating the sufficient condition of adaptivity into the system, firms will attract users to stay longer and gain a competitive advantage.

### 5.2. Theoretical Implications

Our research contributes to the extant literature in the following ways. First, past studies focused on the moderator or mediator considering the relationship between user perception and behavioral intention in the online field (e.g., [65]). This study further builds a moderated mediation model to provide a holistic perspective of business value creation. Second, the study contributes to the online human behavior theory. This study focuses on the relationship between user perception and

stickiness in business microblog contexts and uncovers emotional connection as an intermediate factor, which intervenes between user perception and customer stickiness. Previous studies rarely mentioned the importance of emotional connection as a mediator in Internet contexts. The findings show the emotional connection factor links between perceived usefulness and stickiness. It indicates that increased perception of usefulness induces users to access information via the microblog and increased sharing behavior of users, which strengthens their emotional connections that also results in stickiness [26,38,66]. Moreover, the study adopts adaptivity as a moderator that affects the mediating effect of emotional connection. The study argues that providing more personalized functions or contents influence the mediating process. The results show that the relationship between perceived usefulness and customer stickiness can be mediated by emotional connection, but the mediating effect is not moderated by adaptivity. Further, this study finds a contingent view of the nexus of the mediating effect of emotional connection from perceived ease of use to stickiness. Interestingly, the mediating effect is not significant; however, adding adaptivity could moderate that mediating effect. This result expands the work [67] that suggests a moderator is introduced when there is an unexpectedly weak relation among the constructs. From this respect, once researchers find that the main effect of the independent variable on the dependent variable produces insignificant or inconsistent results, they could consider the moderator to explain such an ambiguous phenomenon (e.g., [68,69]). Namely, that the relationship between the independent variable and the dependent variable may be contingent on the moderator. Our mediating effect of emotional connection from the perceived ease of use to stickiness provides the weak mediating relation, but the moderator of adaptivity has a strong impact on such relation. This implies that when the mediation effects are non-significant or weak, researchers should consider the moderator and argue that the mediating process varies due to the moderator's functions.

## 5.3. Managerial Implications

Our empirical finding about the relationship between user perception and customer stickiness and the moderated mediation model of the microblog has strategic implications for both the microblog service providers and corporations. First, based on our finding, emotional connection mediates the effect of perceived usefulness on customer stickiness. Thus, while operating microblogs, service providers and corporations should offer useful avenues for users so as to increase their motivation to interact with other community members and develop social connection. As the social connection builds, community members may create their own mutual history and then stay longer. Such networks would have a strong tie and might not be replaced by competitors. Second, our results suggest that the mediating effect of emotional connection between perceived ease of use and customer stickiness is moderated by adaptivity. Thus, our study suggests that providing ease of use in the platform of microblog is not enough to attract users to share information and connect with others or even to encourage customers' intention to stay. However, with a higher level of adaptivity, users are more likely to interact with the community and strengthen the mediation effect of emotional connection. Managers could use the data to analyze the user's profile and customize the message for users, thereby attracting user attention and enjoyment in the shopping process.

## 5.4. Limitation and Future Research

This study has some limitations that provide opportunities for further research. First, this study applied emotional connection as the mediating factor to link user perception and stickiness. Future studies can use different psychological factors such as social capital and social ties to discuss such mechanisms. Second, the mediating effect of emotional connection could vary with community adaptivity; however, community adaptivity is not the only factor that moderates this relationship. This study suggests further exploration of additional factors that trigger psychological mechanisms, such as social contagion and users' ability. Third, our study provides the framework under the context of microblogs. This study believes such a theoretical framework could apply to the mobile

shopping context as well, and thus, should be explored. Finally, although the data are reliable and the sample is representative of the population, this study uses cross-sectional data and cannot show causal relationships. Therefore, future research could utilize a longitudinal design to address this issue.

*5.5. Brief Summary*

This study reveals the mediating role of emotional connection on the perceived usefulness and customer stickiness; however, the adaptivity system does not influence this mediating mechanism. Alternatively, this study provides the contingency perspective and indicates that the mediating effect of emotional connection from perceived ease of use to customer stickiness was contingent upon on adaptivity.

**Author Contributions:** Conceptualization: C.-L.H.; formal analysis and methodology: Y.-C.L.; writing: writing—original draft preparation: C.-L.H.; writing—review and editing: Y.-C.L.

**Funding:** This research was funded by Ministry of Science and Technology, R.O.C (Taiwan), Grant Number: 105-2410-H-147-008.

**Acknowledgments:** C.-L.H. is responsible for generating ideas, developing the main framework and hypotheses, and in charge the structure of manuscript; Y.-C.L. is responsible for theorizing the hypotheses, data analysis, and manuscript revisions.

**Conflicts of Interest:** The authors declare no conflict of interest.

## Appendix A. Confirmatory Factor Analysis

| Measure and Source | Operational Measures of Construct | SFL | t-Value |
|---|---|---|---|
| Perceived usefulness ([10] Moon and Kim, 2001); CR = 0.92 | By using this microblog, my purpose can be reached. | 0.81 | 22.44 |
| | By using this microblog, I can make my life more convenient. | 0.85 | 24.06 |
| | By using this microblog, I can make my life more efficient. | 0.81 | 22.53 |
| | By using this microblog, I can obtain more information. | 0.85 | 24.02 |
| | By using this microblog, I can access the latest information. | 0.85 | 24.26 |
| Perceived ease of use ([10] Moon and Kim, 2001); CR = 0.86 | It is easy for me to learn to use this microblog | 0.82 | 21.94 |
| | The interaction between me and this microblog is specific and comprehensible | 0.85 | 230.5 |
| | Mastering the functions in this microblog is easy for me | 0.79 | 21.09 |
| adaptivity ([16] Teo et al., 2003); CR = 0.77 | The microblog I am currently using provides information content according to users' needs | 0.55 | 12.62 |
| | The microblog I am currently using takes the initiative in finding out customers' special requests | 0.85 | 21.26 |
| | The microblog adjust information they provided based on users' needs any time | 0.77 | 18.93 |
| Emotion connection ([4] Hsu and Liao, 2014); CR = 0.90 | I believe the time spent on this micro-blog is worthwhile | 0.82 | 22.55 |
| | I can get what I want from this website. | 0.80 | 21.89 |
| | What I want is similar to what other members of this website want. | 0.82 | 22.84 |
| | The members of this micro-blogging website solve problems together. | 0.79 | 21.42 |
| | The members of this micro-blogging website get alone very well. | 0.77 | 20.57 |
| Stickiness ([53] Liu and Xu, 2010) CR = 0.91 | I think it takes a lot of time and efforts to create a new account in other similar websites. | 0.81 | 22.46 |
| | The cost of time, money, and efforts is high for me to change the micro-blogging website I am using. | 0.88 | 25.34 |
| | I don't want to move to a similar micro-blogging website because I am already familiar with the system of this website. | 0.88 | 25.57 |
| | It's not worthy to take the risk moving to another micro-blogging website. | 0.77 | 20.92 |
| | If I cannot use this micro-blogging website anymore, it'd be a big pity. | 0.71 | 18.67 |

NOTE1: SFL: Standardized Factor Loadings; NOTE2: Model Fit Indices: $\chi2/DF$ = 4.48, CFI = 0.97, IFI = 0.97, RFI = 0.96

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
