# Peer review of "Bridging User Perception and Stickiness in Business Microblog Contexts: A Moderated Mediation Model"

_futureinternet, doi:10.3390/fi11060134_

Round 1

Reviewer 1 Report

Here is my comment according to the 

The major finding is not connect to the practical agenda from business aspect. "The results showed that the relationship between perceived ease of use and customer stickiness is not mediated by emotional connection; however, after considering the moderating effect, our results show that moderated mediation exists." Therefore, what is significant to the actually?

Total 543 sample is used. The statistical soundness is Z score, It would be better to present p-value overall. 

Minor: Instead of We and I, the paper or this study is more appropriate. 

 The literature review research findings are not very easy to read from a complicated table. It would be reader-friendly to sketch a figure of assumed and actual empirical findings.

English Proofreading is needed & Capital letter check should be necessary. 

Author Response

Dear Reviewer,

We would like to express our appreciation for your thorough review and insightful comments. We have made substantial changes to our manuscript based on your suggestions. Below are your comments and our responses (the latter in green) listed point by point. We believe that these comments and suggestions have helped us greatly improve our paper. Thanks again.

Best regards

(1)The major finding is not connect to the practical agenda from business aspect. "The results showed that the relationship between perceived ease of use and customer stickiness is not mediated by emotional connection; however, after considering the moderating effect, our results show that moderated mediation exists." Therefore, what is significant to the actually?

Thank you for this valuable feedback. In this current manuscript, the authors proposed that emotional connection mediates the relationship between user perceptions and customer stickiness, and that this mechanism was contingent on an adaptivity system. The reasons why the authors assert the effect was significant is based on previous research. Specifically, Baron and Kenny (1986) suggested, their famous work, that a moderator is introduced when there is an unexpectedly weak relation among the constructs. With regard to this, as researchers find that the main effect of the independent variable on the dependent variable produces insignificant or inconsistent results, they could consider the moderator to explain such an ambiguous phenomenon. Namely, the relationship between the independent variable and the dependent variable may be contingent on the moderator. For example, Tsai (2009) and Wang et al. (2011) found that past empirical research examining the relationship between the independent variable and the dependent variable was non-significant or inconsistent; therefore, they developed the contingency perspective to argue that the main effect depended on the moderator and offered the reasoning why the main effect was non-significant or inconsistent. Similarly, this study suggested that when the mediation effects are non-significant or weak, researchers could consider the moderator, and demonstrated that the mediating process varies due to the moderator’s functions. For these reasons, even the results of the mediation effect were non-significant. This study further proposed the adaptivity as a moderator to affect the mediating effect of emotional connection from the perceived ease of use to stickiness. The findings denoted that as adaptivity escalates, the mediating effect of emotional connection increases and becomes positively significant. On the contrary, once adaptivity decreases, the mediating effect of emotional connection becomes negatively significant. As such, we state that after considering the moderating effect, our results show that moderated mediation exists.

Please refer to the Discussion section on page 13-14 and the Theoretical Implications section on page 14 for more details.

Reference

Baron, R. and Kenny, D. A. (1986), “The Moderator–Mediator Variable Distinction in Social Psychological Research: Conceptual, Strategic, and Statistical Considerations,” Journal of Personality and Social Psychology, 51, 6, 1173–82.

Tsai, K-H. (2009), “Collaborative networks and product innovation performance: Toward a contingency perspective,” Research Policy, 38, 765-778.

Wang, H. and Qian, C. (2011), “Corporate philanthropy and corporate financial performance: The roles of stakeholder response and political access,” Academy of Management Journal, 54, 6, 1159-1181.

(2)Total 543 sample is used. The statistical soundness is Z score, It would be better to present p-value overall. 

Thank you for this useful suggestion. We added the p-value in our tables. Please refer to Table 2 – Table 5.

(3) Minor: Instead of We and I, the paper or this study is more appropriate. 

Thank you for your suggestion. We have changed “we” to “this study” or “the paper” in this manuscript.

(4) The literature review research findings are not very easy to read from a complicated table. It would be reader-friendly to sketch a figure of assumed and actual empirical findings.

Thank you very much for this useful suggestion. We have presented Figure 2 - 3 for a better understanding of our moderated mediation effects.

(5) English Proofreading is needed & Capital letter check should be necessary. 

Thank you for your advice. We have made sure that the document is edited adequately.

Reviewer 2 Report

The work proposed by the authors, titled “Bridging User Perception and Stickiness in Business Microblog Contexts: A Moderated Mediation Model”, is focused on a study aimed to develop a contingent mediation model to investigate whether user perception enhances customer stickiness through emotional connection and further assess such mediating effect varies with different adaptivity.
In such a context they analyzed the activity related to 543 users from on-line websites, applying a moderated mediation approach in order to test the hypotheses.
They claim that the obtained results prove the mediating role of emotional connection on the link perceived usefulness and customer stickiness, but not moderated by adaptivity.
In addition, they claim that such results show that the relationship between perceived ease of use and customer stickiness is not mediated by emotional connection (but after considering the moderating effect, their results show that a moderated mediation is present).
The scientific contribution declared by the authors is a study where the emotional connection as the mediating factor have been investigated in order to link user perception and stickiness.

I consider that the overall structure of the manuscript is well organized /balanced and the manuscript has been well written.
In any case, I suggest to the authors a careful re-reading of the entire manuscript in order to fix some minor typos (e.g., "Internet" instead of "internet", "all paris of constructs", and so on), and to rewrite some long sentences in a more concise and understandable way.

I also suggest to the authors to avoid in the Abstract forms such as “Purpose--”, “Design/methodology/approach--”, and so on,  adopting instead a discursive and continuous style.

The Introduction section is adequate and it is able to introduce the readers to the research context taken into account by the authors, but in order to better present their scientific contribution I suggest them to present in a list form (e.g., by using a bulleted list).

The formal approach adopted by the authors, as well as the experimental process, have been presented in an understandable form to the readers.
In addition, the used metrics are suitable and the evaluation process has been performed in a correct and reasonable way.

The manuscript needs a Conclusion section, where the authors should summarize the information given in the entire manuscript,  offering a brief but complete summary of the work carried out by the authors, plus a synthetic version of the final discussion.

The references are appropriate and quite updated, but they should include other works centered/close to the domain taken into consideration, in order to offer an overview of the considered scenario to the readers, such as:
(1) Boratto, L., & Carta, S. (2014, June). Modeling the preferences of a group of users detected by clustering: A group recommendation case-study. In Proceedings of the 4th International Conference on Web Intelligence, Mining and Semantics (WIMS14) (p. 16). ACM;
(2) Wang, W. T., Wang, Y. S., & Liu, E. R. (2016). The stickiness intention of group-buying websites: The integration of the commitment–trust theory and e-commerce success model. Information & Management, 53(5), 625-642;
(3) Saia, R., Boratto, L., & Carta, S. (2015, June). A latent semantic pattern recognition strategy for an untrivial targeted advertising. In 2015 IEEE International Congress on Big Data (pp. 491-498). IEEE;
(4) Kalloubi, F., & Nfaoui, E. H. (2016). Microblog semantic context retrieval system based on linked open data and graph-based theory. Expert Systems with Applications, 53, 138-148;
(5) Saia, R., Boratto, L., Carta, S., & Fenu, G. (2016). Binary sieves: Toward a semantic approach to user segmentation for behavioral targeting. Future Generation Computer Systems, 64, 186-197;
(6) Wang, W. T., & Hou, Y. P. (2015). Motivations of employees’ knowledge sharing behaviors: A self-determination perspective. Information and Organization, 25(1), 1-26;
(7) Wang, W. T., & Chang, W. H. (2014). A study of virtual product consumption from the expectancy disconfirmation and symbolic consumption perspectives. Information Systems Frontiers, 16(5), 887-908;
(8) Zhang, M., Guo, L., Hu, M., & Liu, W. (2017). Influence of customer engagement with company social networks on stickiness: Mediating effect of customer value creation. International Journal of Information Management, 37(3), 229-240;
(9) Liang, T. P., Chen, H. Y., & Turban, E. (2009, August). Effect of personalization on the perceived usefulness of online customer services: A dual-core theory. In Proceedings of the 11th International Conference on Electronic Commerce (pp. 279-288). ACM.

In conclusion, up to my knowledge, the scientific contribution given by the proposed work can be consider valuable in the research field taken into consideration by the authors and, in addition, it well fits the scope of the Future Internet journal, although before its publication the manuscript needs to be revised, according to the aforementioned observations.

Author Response

Dear Reviewer,

Thank you for providing us your valuable and insightful comments. We have made revisions based on your suggestions. Below are your comments and our responses listed point by point (the latter in purple). Your comments have helped us significantly improve our paper. Thanks again.

Best regards

The work proposed by the authors, titled “Bridging User Perception and Stickiness in Business Microblog Contexts: A Moderated Mediation Model”, is focused on a study aimed to develop a contingent mediation model to investigate whether user perception enhances customer stickiness through emotional connection and further assess such mediating effect varies with different adaptivity. In such a context they analyzed the activity related to 543 users from on-line websites, applying a moderated mediation approach in order to test the hypotheses.
They claim that the obtained results prove the mediating role of emotional connection on the link perceived usefulness and customer stickiness, but not moderated by adaptivity.
In addition, they claim that such results show that the relationship between perceived ease of use and customer stickiness is not mediated by emotional connection (but after considering the moderating effect, their results show that a moderated mediation is present).
The scientific contribution declared by the authors is a study where the emotional connection as the mediating factor have been investigated in order to link user perception and stickiness.

(1) I consider that the overall structure of the manuscript is well organized /balanced and the manuscript has been well written.

Thank you for your encouragement and your assistance in improving the manuscript.

(2) In any case, I suggest to the authors a careful re-reading of the entire manuscript in order to fix some minor typos (e.g., "Internet" instead of "internet", "all paris of constructs", and so on), and to rewrite some long sentences in a more concise and understandable way.

We have corrected these typos. In addition, we have also checked and            corrected other issues throughout the text. Moreover, we have rewritten some sentences more succinctly to present our argument.

(3) I also suggest to the authors to avoid in the Abstract forms such as “Purpose--”, “Design/methodology/approach--”, and so on, adopting instead a discursive and continuous style.

Thank you for your suggestion. We have revised our abstract for better flow and clarity.

(4) The Introduction section is adequate and it is able to introduce the readers to the research context taken into account by the authors, but in order to better present their scientific contribution I suggest them to present in a list form (e.g., by using a bulleted list).

Thank you for this vital feedback. We have presented the information in a list form to summarize our contribution at the end of our Introduction section.

(5) The formal approach adopted by the authors, as well as the experimental process, have been presented in an understandable form to the readers. In addition, the used metrics are suitable and the evaluation process has been performed in a correct and reasonable way.

Thank you very much for your encouragement.

(6) The manuscript needs a Conclusion section, where the authors should summarize the information given in the entire manuscript, offering a brief but complete summary of the work carried out by the authors, plus a synthetic version of the final discussion.

Thank you for this suggestion. We followed your recommendation and adjusted our Conclusion section. This section includes the Discussion, Theoretical Implications, Managerial Implications, Limitations and Future Research, and Brief Summary. Through this amendment, we provide a more clear and concrete contribution of our research.

(7) The references are appropriate and quite updated, but they should include other works centered/close to the domain taken into consideration, in order to offer an overview of the considered scenario to the readers, such as:
(7-1) Boratto, L., & Carta, S. (2014, June). Modeling the preferences of a group of users detected by clustering: A group recommendation case-study. In Proceedings of the 4th International Conference on Web Intelligence, Mining and Semantics (WIMS14) (p. 16). ACM;
(7-2) Wang, W. T., Wang, Y. S., & Liu, E. R. (2016). The stickiness intention of group-buying websites: The integration of the commitment–trust theory and e-commerce success model. Information & Management, 53(5), 625-642;
(7-3) Saia, R., Boratto, L., & Carta, S. (2015, June). A latent semantic pattern recognition strategy for an untrivial targeted advertising. In 2015 IEEE International Congress on Big Data (pp. 491-498). IEEE;
(7-4) Kalloubi, F., & Nfaoui, E. H. (2016). Microblog semantic context retrieval system based on linked open data and graph-based theory. Expert Systems with Applications, 53, 138-148;
(7-5) Saia, R., Boratto, L., Carta, S., & Fenu, G. (2016). Binary sieves: Toward a semantic approach to user segmentation for behavioral targeting. Future Generation Computer Systems, 64, 186-197;
(7-6) Wang, W. T., & Hou, Y. P. (2015). Motivations of employees’ knowledge sharing behaviors: A self-determination perspective. Information and Organization, 25(1), 1-26;
(7-7) Wang, W. T., & Chang, W. H. (2014). A study of virtual product consumption from the expectancy disconfirmation and symbolic consumption perspectives. Information Systems Frontiers, 16(5), 887-908;
(7-8) Zhang, M., Guo, L., Hu, M., & Liu, W. (2017). Influence of customer engagement with company social networks on stickiness: Mediating effect of customer value creation. International Journal of Information Management, 37(3), 229-240;
(7-9) Liang, T. P., Chen, H. Y., & Turban, E. (2009, August). Effect of personalization on the perceived usefulness of online customer services: A dual-core theory. In Proceedings of the 11th International Conference on Electronic Commerce (pp. 279-288). ACM.

Thank you for providing these references to help us improve the manuscript. We have studied these articles and have referenced some of them.

(9) In conclusion, up to my knowledge, the scientific contribution given by the proposed work can be consider valuable in the research field taken into consideration by the authors and, in addition, it well fits the scope of the Future Internet journal, although before its publication the manuscript needs to be revised, according to the aforementioned observations.

Thank you for your careful consideration and your insightful review. We have revised our manuscript in response to your concerns.

Round 2

Reviewer 1 Report

The paper presents an analysis on a contingent mediation model to investigate whether user perception enhances customer stickiness through emotional connection and further assess such mediating effect varies with different adaptivity. Overall research methodology and revised manuscript is satisfactory.